# OpenReview forum: "Output Scouting: Auditing Large Language Models for Catastrophic Responses"
_ICLR.cc/2025/Conference — ICLR 2025 Conference Withdrawn Submission_

### Official Review · Reviewer_ytix · 2024-10-27

**Soundness:** 2
**Presentation:** 2
**Contribution:** 3
**Rating:** 3
**Confidence:** 4

**Summary:**

This paper focuses on the issue of **triggering and detecting catastrophic responses** in large language models. More specifically, the problem refers to how to design a strategy to find catastrophic responses efficiently, within a limited query time (e.g., 1000). To address this issue, the authors propose a temperature adjustment strategy called **OUTPUT SCOUTING**, which can **generate semantically fluent outputs for a given prompt that match any target probability distribution**. Based on this technique, the authors conducted experiments on six different prompts (query 1000 times for each), and through human evaluations, they confirmed that OUTPUT SCOUTING has significant advantages compared to vanilla sampling (temperature = 0.5). The authors also promise to release the related toolkit.

**Strengths:**

1. The topic is critical and interesting. A way to effectively detect or warn about catastrophic responses could reduce harm to users when using LLMs, and also lessen the potential losses for the organizations that own these models.
2. The method in the paper is lightweight and easy to implement.
3. The authors promise that they will release a toolkit regarding the auditing framework.

**Weaknesses:**

1. I think the biggest concern with this paper is that the experiments lack comprehensiveness and soundness：
- **Only 6 prompts were used in the experiment**: In the experiment, only 6 queries were used (see Table 1), and the sentences were very similar (e.g., "Can I xxx?"). This makes me doubt whether the method would still work well in different scenarios. Although using more evaluation queries would increase the cost—since each model generates 1000 responses for each query, and these responses need to be evaluated manually—I believe it is necessary to have a larger and more diverse set of queries.
- **The experiment lacks soundness**: The author compared vanilla sampling with temperature = 0.5 in Table 1, and the results show that output scouting (temperature range from 0 to 10.0) can find significantly more catastrophic responses. For example, in the first case, output scouting found 28 such responses from Llama3-8B-Instruct, while vanilla sampling found only 6. My question is: Since increasing the temperature will raise randomness, can adjusting the vanilla model's temperature achieve results comparable to output scouting? I think adding relevant experiments is necessary (more different hyperparameters and more common sampling strategies) to ensure the soundness of the study.
2. Although detecting and identifying catastrophic responses is very important, it is not enough. This paper lacks a discussion on possible solutions. This makes the paper seem incomplete.

**Questions:**

1. Could you explain what direct insights we can get from using output scouting to find catastrophic responses on how to solve these issues? For example, if we find that the model has a catastrophic response with a very low normalized probability, what should we do next? Should we perform safety tuning for this specific query?
2. Could you please explain your insight in more detail? Specifically, regarding how your approach identifies catastrophic responses, what types are easily detected by your method, and which types are more likely to be missed?

---

### Official Review · Reviewer_Zqp3 · 2024-11-03

**Soundness:** 2
**Presentation:** 4
**Contribution:** 2
**Rating:** 5
**Confidence:** 4

**Summary:**

The paper introduces a novel approach for auditing harmful outputs of LLMs by adapting the sampling process to find a specific type of output given a target probability distribution as efficiently as possible. After the problem statement and the introduction of the proposed method, the authors evaluate two open-weights LLMs on six example prompts which are based on popular benchmarks. The introduced output scouting approach is compared to vanilla sampling.

**Strengths:**

- A novel approach to probe LLM’s likelihood of generating potential harmful responses, assisting human evaluation.
The problem is well-motivated and clearly described. The proposed method is well-introduced.

- The proposed approach will be provided as an open-source toolkit

- Limitations are transparently described.

**Weaknesses:**

- Practical impact unclear. As the authors state, the proposed approach does not guarantee semantically fluent outputs. Unfortunately, the paper does not provide an analysis in this regard. Further, LLMs are usually not solely deployed in an application without any safeguards. For example, Meta advises deploying their llama models along with their LlamaGuard models for input and output moderation. See the level safety section in https://ai.meta.com/blog/llama-3-2-connect-2024-vision-edge-mobile-devices/
A safety inspector, the target application of the present paper, would therefore most unlikely only audit an LLM but rather the whole AI system, such as a chatbot application.

- Missing details on the models’ (harmful) responses and human evaluation. The authors argue that the detected outputs are fluent. Providing examples or other evidence would strengthen this statement. Additionally, the human evaluation lacks clarity, including whether the evaluators were external or part of the author team, their qualifications, and the criteria used for evaluation.

- It is unclear if the approach will scale since the evaluation is based only on 6 example prompts. Further clarification on the computational requirements for applying this approach to larger safety benchmarks, such as ALERT (https://arxiv.org/abs/2404.08676), would provide valuable insight into its scalability

- The authors recommend using human evaluators since LLM-based guardrails to detect unsafe responses are vulnerable to adversarial attacks. However, this seems not to be relevant in the context of the present study.


**Minor Comments:**
- In Section 3.1, I assume you intended to refer to Figure 1 instead of Figure 3.

- Inconsistent citing style and use of fields such as url, pages, etc.

**Questions:**

- Have you conducted an assessment using LLM-based guardrails to detect unsafe responses? Since the authors argue against using LLM guardrails, this would further strengthen their argument.

- Can you provide examples of the extracted responses?

- Can you elaborate on how output scouting could be integrated into auditing a complete AI system with safeguard integration?

---

### Official Review · Reviewer_7zen · 2024-11-04

**Soundness:** 3
**Presentation:** 3
**Contribution:** 2
**Rating:** 6
**Confidence:** 4

**Summary:**

This paper addresses an important challenge of auditing LLMs to identify potentially harmful or catastrophic responses that could emerge when these models are deployed on a large scale.

The main contributions of the paper are as follows:

- Output Scouting: An efficient strategy that leverages an auxiliary temperature parameter to query LLMs and generate a set of outputs that align with any specified probability distribution.
- Framework for Auditing: A comprehensive framework around output scouting designed to audit LLMs for harmful or catastrophic responses.
- Experimental Results: Tests conducted on Meta-Llama-3-8B-Instruct and Mistral-7B-Instruct-v0.3 demonstrate that the proposed method identifies significantly more harmful responses compared to traditional sampling techniques.

Additionally, the authors propose to release an open-source toolkit implementing the framework described.

**Strengths:**

- The paper introduces a simple and novel strategy for generating responses from any LLM that matches a specified probability distribution, offering an improvement over the uniform distribution generated by vanilla sampling.
- The paper is well-written overall. The motivation for the work and the details of the proposed framework are clearly explained.

**Weaknesses:**

- As acknowledged by the authors, conducting a complete audit of an LLM as suggested in the paper is challenging, as the number of prompts or questions required can be very large.
- Also, the output-scouting-based auditing approach relies on human evaluators to assess all sampled responses, which is labor-intensive and may not be scalable for real-world applications.
- Experimental results are limited, with evaluations performed on only six prompts and two language models.

**Questions:**

- For vanilla sampling, does using higher temperature parameters, such as 1.0 or 1.5, help identify more harmful responses?
- For output scouting, what fraction of generated responses contains gibberish text (even if only partially gibberish)?

---

### Note · Authors · 2024-11-19

I have read and agree with the venue's withdrawal policy on behalf of myself and my co-authors.